# HER2-Low Luminal Breast Carcinoma Is Not a Homogenous Clinicopathological and Molecular Entity

**DOI:** 10.3390/cancers16112009

**Published:** 2024-05-25

**Authors:** Céline André, Aurélie Bertaut, Sylvain Ladoire, Isabelle Desmoulins, Clémentine Jankowski, Françoise Beltjens, Céline Charon-Barra, Anthony Bergeron, Corentin Richard, Romain Boidot, Laurent Arnould

**Affiliations:** 1Unit of Pathology, Department of Tumor Biology and Pathology, Georges-François Leclerc Cancer Center, 21000 Dijon, France; f.beltjens@cgfl.fr (F.B.); c.charonbarra@cgfl.fr (C.C.-B.); a.bergeron@cgfl.fr (A.B.); l.arnould@cgfl.fr (L.A.); 2Unit of Pathology, University Hospital Center, 21000 Dijon, France; 3Unit of Methodology and Biostatistics, Georges-François Leclerc Cancer Center, 21000 Dijon, France; a.bertaut@cgfl.fr; 4Department of Medical Oncology, Georges-François Leclerc Cancer Center, 21000 Dijon, France; s.ladoire@cgfl.fr (S.L.); i.desmoulins@cgfl.fr (I.D.); 5Unit 1231 (INSERM U1231), National Institute of Health and Medical Research, 21000 Dijon, France; 6Department of Medicine, University of Burgundy Franche-Comté, 21000 Dijon, France; 7Department of Surgery, Georges-François Leclerc Cancer Center, 21000 Dijon, France; c.jankowski@cgfl.fr; 8Unit of Molecular Pathology, Department of Tumor Biology and Pathology, Georges-François Leclerc Cancer Center, 21000 Dijon, France; c.richard@cgfl.fr (C.R.); r.boidot@cgfl.fr (R.B.)

**Keywords:** breast carcinomas, HER2-low, *PIK3CA* mutation

## Abstract

**Simple Summary:**

The new HER2-low category, comprising HER2 IHC 1+ and 2+ carcinomas, expressing predominantly hormone receptors, has been added to the HER2 classification of breast carcinoma. With the advent of antibody–drug conjugates, these carcinomas need to be better characterized at the clinicopathological, molecular, and transcriptomic levels. We analyzed 62 HER2-low luminal carcinomas, comparing them with 43 HER2-positive and 20 HER2-negative carcinomas. The transcriptomic activities of three HER2 effector pathways (PI3K-AKT, MAPK, and JAK-STAT) were investigated using RNA sequencing, and the mutational status of key breast cancer-associated genes was determined using DNA sequencing. The impact of the presence of a *PIK3CA* mutation appears to be essential in the activation of the PI3K-AKT signaling pathway. *PIK3CA* mutations could be a lead in variable responses to conventional anti-HER2 therapies.

**Abstract:**

Background: With the development of some new antibody–drug conjugates, the HER2 classification of breast carcinomas now includes the HER2-low (H2L) category: IHC 1+, 2+ non-amplified by ISH, and double-equivocal carcinomas, mostly luminal, expressing hormone receptors (HR+). Methods: We analyzed mutational status and transcriptomic activities of three HER2 effector pathways: PI3K-AKT, MAPK, and JAK-STAT, in association with clinicopathologic features, in 62 H2L carcinomas compared to 43 HER2-positive and 20 HER2-negative carcinomas, all HR+. Results: H2L carcinomas had significantly lower histoprognostic grades and mitotic and Ki67 proliferation indexes than HER2-positive carcinomas. Their *PIK3CA* mutation rates were close to those of HER2-negative and significantly higher than in HER2-positive carcinomas, contrary to *TP53* mutations. At the transcriptomic level, we identified three distinct groups which did not reflect the new HER2 classification. H2L and HER2-negative carcinomas shared most of clinicopathological and molecular characteristics, except HER2 membrane expression (mRNA levels). The presence of a mutation in a signaling pathway had a strong pathway activation effect. *PIK3CA* mutations were more prevalent in H2L carcinomas, leading to a strong activation of the PI3K-AKT signaling pathway even in the absence of HER2 overexpression/amplification. Conclusion: *PIK3CA* mutations may explain the failure of conventional anti-HER2 treatments, suggesting that new antibody–drug conjugates may be more effective.

## 1. Introduction

Breast carcinomas are the most common female malignancy. About two million new cases were diagnosed in 2022, making it the fourth leading cause of cancer-related death in women worldwide. Between 15 and 20% of primary invasive breast carcinomas show overexpression and/or amplification of the Human Epithelial Growth Factor Receptor-2 (HER2), a transmembrane glycoprotein with tyrosine-kinase activity encoded by the *ERBB2* gene on chromosome 17 (17q12).

Determining the HER2 status is central to the therapeutic management of breast carcinoma patients. It requires standardized immunohistochemistry (IHC) testing and result interpretation to assess the protein expression level, and additional testing by an in situ hybridization (ISH) for gene status assessment in case of 2+ IHC staining. In current clinical practice, the HER2 status assessment follows the 2018 update of the recommendations by the American Society of Clinical Oncology and the College of American Pathologists (ASCO/CAP) [1,2,3] introducing five ISH groups that emphasize concomitant interpretation of both techniques. In addition, a category of HER2-low (H2L) has recently emerged. It is assigned to carcinomas with an IHC assay score of 1+ or 2+ but with a negative ISH result (including ASCO/CAP group 4), and represents 45–55% of all breast carcinomas [4,5]. In clinical practice, these carcinomas used to be defined as HER2-negative, although they express some levels of HER2 on the cell membrane, detectable with IHC.

The need for a precise HER2 status classification is also linked to a major revolution in the management of breast carcinoma patients due to the introduction of treatments that substantially improve patient outcomes. These include Trastuzumab, an anti-HER2 targeted therapy, whose effectiveness was described in metastatic [6], then adjuvant [7,8], and neoadjuvant [9] settings. However, since the 2018 update of the ASCO/CAP Guidelines, patients with group 4 carcinomas are no longer eligible for this treatment due to lack of sufficient clinical benefit.

The tumor HER2 status has been linked to the probability of achieving complete pathological response (pCR) to an anti-HER2 treatment (Trastuzumab or double blocking with Trastuzumab-Pertuzumab) in a neo-adjuvant setting, with the HER2/CEP17 ratio having a predictive value on its own: patients with weakly amplified carcinomas have low rates of pCR [10,11].

Some hope to improve the outcomes of HER2-positive and H2L has come with the emergence of new antibody–drug conjugates (ADCs). Among those, Trastuzumab-Deruxtecan (T-DXd, DS8201a), an anti-HER2 antibody linked to a topoisomerase I inhibitor, is now routinely prescribed to metastatic patients after failure of one or two lines of chemotherapy, since it has been shown to be associated with a significantly longer progression-free survival and overall survival than chemotherapy [12,13]. Therefore, an in-depth characterization of weakly amplified breast carcinomas, as well as all breast carcinomas with regard to different levels of HER2 expression, has become one of clinical priority.

Still, the HER2 status alone is not sufficient to predict tumor behavior and guide the choice of the most effective targeted therapies. It is therefore essential to investigate other molecular alterations and their interconnections in order to understand the underlying oncogenic mechanisms and inform treatment decisions. The HER2 receptor, like most receptor tyrosine kinases, influences three major intracellular signaling pathways, all involved in tumor survival and proliferation: the PI3K-AKT pathway (activated by HER2 dimerization at the cell surface and impacted by mutations in the *PIK3CA*, *AKT1*, and *PTEN* genes), the MAP-kinase (MAPK) pathway (a kinase cascade with signal amplification at each new phosphorylation step and impacted by *KRAS*, *NRAS*, and *BRAF* mutations), and the JAK-STAT pathway (an independent pathway not affected by mutations in genes affecting the other two pathways). The PI3K-AKT and MAPK pathways are inter-connected, and both are affected by mutations in genes encoding actors involved in either of the two, in particular, mutations in the *PIK3CA* gene (the p110α subunit of PI3K upregulates the MAPK pathway, and the RAS protein activates the PI3K protein by potentiating its lipid kinase activity) [14,15].

In this study, we aimed to describe the clinicopathological, molecular, and transcriptomic profiles of invasive H2L breast carcinomas compared to HER2-negative and HER2-positive carcinomas to determine whether some differences between these tumor categories could explain the known differences in response to treatments. The secondary objective was to analyze the same parameters for patients with the ASCO/CAP group 4 carcinomas, a subcategory of the H2L carcinomas for which the available data, and especially molecular data, remain particularly scarce, and compare them to the characteristics of the other HER2 categories.

## 2. Materials and Methods

### 2.1. Study Design, Patients, and Samples

This study was comprised of 125 breast carcinoma patients, including 62 patients with H2L carcinomas, 20 patients with HER2-negative carcinomas, and 43 patients with HER2-positive carcinomas.

Due to low prevalence of primary invasive breast carcinomas meeting the criteria of the ASCO/CAP group 4, we started by including all patients with these carcinomas diagnosed between January 2018 and December 2020 at the Georges-François Leclerc Cancer Center (CGFL; Dijon, France) on surgical specimens (*n* = 22). All of them had carcinomas which expressed both progesterone and estrogen receptors (PR and ER, i.e., hormone-receptor-positive (HR+) carcinomas), and received hormone therapy, but not an anti-HER2 targeted treatment. These carcinomas are referred to as DE (for “double-equivocal”) in this article.

Then, we randomly selected 20 HER2 IHC 1+ carcinomas, and 20 HER2 IHC 2+ carcinomas without HER2 amplification by ISH (further called 2+ NA for “non-amplified”, group 5), for a total of 62 H2L carcinomas. Patients with lobular carcinomas or carcinomas lacking the invasive component (in situ carcinomas), those with a post-treatment recurrence, and patients who received neoadjuvant therapy without available prior biopsy were excluded from the study.

Patients with HER2-negative and HER2-positive carcinomas were randomly selected from the same database among patients diagnosed with an invasive HR+ carcinoma on a surgical specimen (*n* = 54) or a biopsy (*n* = 9, all HER2 3+ neoadjuvant specimens) between January 2007 and December 2020, and having received hormone therapy. Among patients with HER2-positive carcinomas, 20 had weakly amplified (2 + WA) carcinomas (HER2 IHC 2+ with weak amplification signal in ISH, i.e., copy numbers between 6 and 10, whatever the ratio) and 23 had strongly positive carcinomas (HER2 IHC 3+ with more than 10 copy numbers by ISH). All patients with HER2-positive carcinomas received an anti-HER2 therapy. Exclusion criteria were the same as for the other categories, with an additional criterion of HR-negativity (to match with DE carcinomas which were all HR+).

For all study subjects, the demographic and clinical data were collected retrospectively from computerized medical records including ultrasound scan reports with tumor size (T, multifocality) and lymph node status (N) classified according to the current TNM staging criteria defined by the American Joint Committee on Cancer Staging classification (AJCC, 8th edition) [16].

The CGFL was authorized to conduct this study by relevant French authorities (authorization number AC-2019-3531). The study was approved by the CGFL Ethical and Scientific Committee.

### 2.2. Histopathological Review

Histopathological data were directly extracted from histologic reports. For patients who received neo-adjuvant therapy (*n* = 9; all had HER2 3+ carcinomas), only data concerning the initial pre-treatment biopsy were analyzed. The histologic subtypes were defined according to the Fifth Edition of the World Health Organization’s Classification of Breast Tumors [17]. The Bloom and Richardson grading system modified by Elston and Ellis (E&E histologic grading) [18] was used to characterize the carcinoma, including mitosis score supplemented using the mitotic index (/mm^2^) and the TNM AJCC classification to determine tumor size and lymph node status on a dissected sentinel or axillary lymph node when pT and pN data were available.

All samples were centralized at the CGFL. In order to confirm diagnosis-related data, all slides (hematoxylin-eosin, Ki67, and HER2 staining) and result images (HER2 ISH results) were reviewed by two pathologists blinded to the tumor HER2 category. A semi-quantitative evaluation of tumor-infiltrating lymphocyte (TILs) levels was also performed by two pathologists following the 2014 recommendations of the International TILs Working Group [19]. The results were recorded as percentages of lymphocytes within the tumor bed.

### 2.3. ER, PR, and Ki67 Protein Expression Analysis Using Immunohistochemistry (IHC)

The expression of estrogen and progesterone receptors (ER and PR) and of the Ki67 proliferation index was analyzed using immunohistochemistry. All analyses were performed on 5 μm thick formalin-fixed paraffin-embedded (FFPE) tissue sections. The protocol details with antibody references are provided in Appendix A. The results were systematically scored by two independent pathologists and classified according to the latest international recommendations (Appendix A). In case of a disagreement, the slides were reviewed by a third pathologist to find a consensus.

Of note, carcinomas with ER or PR scores between 1% and 9% were excluded from our study because there is too little evidence for the efficacy of hormonal therapy in patients with these carcinomas, and because these very weakly positive carcinomas present biological and transcriptomic profiles which are very close to those of ER-negative carcinomas [20].

### 2.4. HER2 Status Assessment

The HER2 status (overexpression and/or amplification) of carcinomas was assessed using IHC and fluorescent in situ hybridization (FISH) according to the latest ASCO/CAP 2018 guidelines [2]. The HER2 expression was first scored using IHC (Clone 4B5, Ventana Benchmark XT system^®^, Roche™, Bale, Switzerland). It was classified as negative in case of 0+ (no staining or low intensity membrane staining in less than 10% tumor cells) and 1+ staining (low intensity membrane staining in more than 10% tumor cells), and positive in case of 3+ scores (high intensity membrane staining >10% tumor cells). Carcinomas with a 2+ IHC score, moderate intensity membrane staining in over 10% tumor cells or strong intensity staining in less than 10% tumor cells were additionally analyzed using FISH to discern between group 2+ WA, ASCO/CAP group 4 (DE), and group 5 (2+ NA) carcinomas.

The FISH analysis was conducted on 4 μm FFPE tissue sections using the dual HER2/CEP17 probe FISH assay using Hybridizer system^®^ with Zytovision kit (ZytoLight^®^ SPEC ERBB2/CEN 17 Dual Color Probe, Clinisciences™, Nanterre, France) according to the manufacturer’s protocol. The results were reviewed by two independent pathologists on at least 40 nuclei. The average HER2 copy number and the HER2/CEP17 ratio were recorded for each carcinoma. The carcinomas were then classified into five groups following the 2018 ASCO/CAP guidelines [2] as follows: group 1 amplified carcinomas (HER2/CEP17 ratio ≥ 2 and ≥4 HER2 signals/cell), group 2 carcinomas (HER2/CEP17 ratio ≥ 2 and <4 HER2 signals/cell), group 3 amplified carcinomas (HER2/CEP17 ratio < 2 and ≥6 HER2 signals/cell), group 4 carcinomas (HER2/CEP17 ratio < 2, and between 4 and 6 HER2 signals/cell), and group 5 non-amplified carcinomas (HER2/CEP17 < 2 and < 4 HER2 signals/cell). Groups 2 and 4 are “non-classical” and require a second blind reading. If the initial result is confirmed, group 4 (formerly called double-equivocal) is classified as “negative with concomitant IHC data”, now referred to as H2L.

### 2.5. Molecular Analysis

#### 2.5.1. DNA and RNA Extraction

Based on the evaluation of the percentage of infiltrating tumor cells on H&E slides, a macrodissection of the samples was performed in order to extract nucleic acids from FFPE specimens. All specimens had at least 60% tumor cells, except for three specimens which had between 40% and 60% tumor cells (all the three were HER2 3+ biopsies). DNA was extracted from four 5 μm tumor slides using Maxwell-16 FFPE Plus LEV DNA purification kit (Promega Corporation, Madison, WI, USA) according to the manufacturer’s protocol. RNA was extracted from the same tumor areas of all specimens using Maxwell-16 LEV RNA FFPE Purification kit (Promega Corporation) according to the manufacturer’s protocol. DNA and RNA quality were assessed using spectrophotometry. DNA was quantified using the Qubit device 4 fluorometric assay (Life Technologies, Thermo Fisher Scientific, Inc., Waltham, MA, USA).

#### 2.5.2. DNA Sequencing

For library preparation, 400ng DNA from carcinomas were fragmented with a Covaris LE220-plus device (Covaris, Inc., Woburn, MA, USA) to obtain fragments ~300 bp-long. Then, libraries were prepared with a SureSelectXT custom panel containing *PIK3CA*, *AKT1*, *PTEN*, *TP53*, *BRCA1*, *BRCA2*, *PALB2*, *ARID1A*, *KRAS*, *NRAS*, and *BRAF* genes (Agilent Technologies, Inc., Santa Clara, CA, USA) following the manufacturer’s instructions. Paired-end (2 × 111 bases) sequencing was performed on a NextSeq500 device (Illumina, Inc., San Diego, CA, USA).

To analyze the results, the reads in the FASTQ format were aligned to the reference human genome GRCh37 using the Burrows–Wheeler aligner (BWA v.0.7.15) as described by others [21]. Local realignment was performed using the Genome Analysis Toolkit (GATK v.3.6) [22,23,24]. Duplicate reads were removed using Picard v.2.5 [25]. Outlyzer (v1.0) [26] was used to identify variants, and the Annovar (Annovar2016Feb01) [27] and SnpEff (v4.3i) [28] tools to annotate them. Quality controls were performed using fastQC (v0.11.8) [29], Samtools (v1.9) [30], and Qualimap (v2.2.1) [31] information through multiQC (v1.7) software [32]. Variants with a frequency above 1% in the general population were filtered out and excluded from result tables before result analysis.

#### 2.5.3. RNA Sequencing and Transcriptomic Analysis

rRNA-depleted RNA was used for the library preparation with the NEBNext Ultra II Directional RNA library prep kit for Illumina (New England Biolabs) according to the manufacturer’s instructions. Libraries were paired-end sequenced (2 × 76 base pairs) on a NextSeq500 device (Illumina), with a read depth of 20 million.

Kallisto software (v0.50.0) was used for quantifying transcript abundance from RNA-seq data against GRCh38 cDNA reference transcriptome from the Ensembl database, v96 [33]. Downstream analysis included only protein-coding genes and transcripts. Differential expression analysis was performed using the DESeq2 R package [34] and single-sample Gene Set Enrichment Analysis using the GSVA R package v1.40.1 [35].

We focused on PI3K-AKT, MAPK, and JAK-STAT intracellular signaling pathways. Only breast cancer-relevant genes downstream of these pathways were retained for the analyses.

### 2.6. Statistical Analysis

Continuous variables were expressed as numbers of observations, means (with standard deviations), and medians (with min–max), and compared using the Student, Wilcoxon, Anova, or Kruskal–Wallis test, as appropriate. Categorical variables, expressed as frequencies and percentages, were compared using the Chi^2^ or Fisher test. If a difference was found, post hoc tests were performed with Bonferroni corrections to identify the groups between which this difference occurred. All tests were two-sided. The significance threshold was set at 5%. All statistical analyses were performed using the SAS software, version 9.4.

## 3. Results

### 3.1. Clinicopathological Characteristics of Patients and Carcinomas

Baseline clinical and histologic characteristics are summarized in Table 1 and Table 2.

We found that H2L carcinomas had lower histologic grades than HER2-positive carcinomas (*p* = 0.0017) with lower mitotic indexes and lower Ki67 expression (*p* = 0.0042 and 0.0003). The same differences appeared between H2L and 2+ WA carcinomas, with lower grades (*p* = 0.036), mitotic scores and indexes (*p* = 0.0357 and 0.01), and lower Ki67 proliferation indexes (*p* = 0.0257) for H2L carcinomas. In contrast, we found no parameter which could differentiate HER2-negative and H2L carcinomas. There were no significant differences between the three carcinoma groups in terms of tumor size, nodal status, the presence of lymphatic vascular emboli, and the levels of tumor-infiltrating lymphocytes (TILs).

When analyzing the six groups, we found significant differences in mitotic scores and indexes between DE with HER2 0+ carcinomas (*p* = 0.0156 and 0.0065). The DE carcinomas had significantly higher Ki67 proliferation indexes compared to 0+ and 1+ carcinomas (*p* = 0.0175 and 0.0011), and higher mitotic indexes than 2+ NA carcinomas (*p* = 0.0189).

### 3.2. Genomic Profiles

We successfully sequenced genes of the PI3K-AKT, JAK-STAT, and the MAPK pathways for 122 of 125 carcinomas. Three samples (all HER2-positive) did not have enough DNA for sequencing. All molecular data are detailed in Table 3 and Table 4. A complete list of pathogenic variants is provided in Appendix A.

Out of all genes analyzed in the three pathways, we found significant differences in the mutation prevalence only for the *PIK3CA* and *TP53* genes. We identified 42 activating (gain-of-function) *PIK3CA* mutations, with the highest prevalence in H2L carcinomas (45.2%). These mutations were significantly more frequent among H2L carcinomas than in the HER2-positive group (*p* = 0.0048). Of note, no significant difference in the *PIK3CA* mutation prevalence was found between H2L and 2+ WA carcinomas, or between HER2-negative and H2L carcinomas. Overall, a higher *PIK3CA* mutation prevalence correlated with the absence of the *ERBB2* gene amplification/hyperexpression (*p* = 0.0063). Among H2L carcinomas, these mutations occurred mostly in grade I carcinomas (*n* = 13; 46.43%), smaller than 2 cm (pT1; *n* = 18; 64.28%) and without axillary node invasion (pN0; *n* = 24; 85.71%). Moreover, the mutant carcinomas had low mitotic counts (71.43% were score 1 carcinomas) and low Ki67 proliferation indexes (median 10%). Lymphovascular emboli were present in six mutant carcinomas (21.43%)—a lower proportion than for all H2L carcinomas (33.9%). TILs levels were low, with a median of 5%. The prevalence of *PIK3CA* mutations in the DE group (36.4%) was almost as high as in all other H2L carcinomas groups (1+ and 2+ NA carcinomas) together (*p* = 0.0227). However, in one-to-one comparisons, we found no significant differences in the *PIK3CA* mutation prevalence between DE carcinomas and any of the other carcinoma groups, including the two HER2-positive carcinoma groups (2+ WA and 3+ carcinomas).

In total, 15 of 122 carcinomas (12%) carried pathogenic variants in the *TP53* gene. We observed an increased mutation prevalence among HER2-positive carcinomas compared to the other two groups (*p* = 0.0003), with 3 mutations found in H2L carcinomas and 12 in HER2-positive carcinomas (*p* = 0.0028). All occurred in carcinomas with high (II and III) histologic grades (66% and 33%) and high Ki67 proliferation indexes (median 20%). All the three H2L mutant carcinomas were classified as HER2 2+ by immunohistochemistry (two 2+ NA and one DE). The *TP53* mutation prevalence in the 2+ WA group did not differ from that among all H2L carcinomas. Very low *TP53* mutation prevalence in the DE group (only one mutation) did not allow us to find a statistically significant association which would distinguish these carcinomas from the other groups. However, we observed a significant link between the *TP53* mutation prevalence and HER2 amplification and/or hyperexpression when comparing the six groups (*p* = 0.0004).

### 3.3. Transcriptomic Profiles

We obtained complete transcriptomic profiles for 120 of 125 carcinomas. The remaining five samples (2 H2L and 3 HER2-positive carcinomas) did not have enough RNA for sequencing.

#### 3.3.1. ERBB2 mRNA Expression

The *ERBB2* mRNA expression levels in H2L carcinomas were significantly lower than those of HER2-positive carcinomas (*p* < 0.0001) and very close (no significant difference) to those of HER2-negative carcinomas (Figure 1). A detailed comparison of the six IHC groups showed an increasing gradient of *ERBB2* expression from the HER2 0+ group to the HER2 3+ group. The latter carcinomas had significantly higher *ERBB2* expression levels than all the other groups (*p* < 0.0001 for the difference with the 0+, 1+, and 2+ NA carcinomas; *p* = 0.0017 for DE and *p* = 0.0022 for 2+ WA carcinomas). The *ERBB2* expression appeared to be similar between HER2 0+ and 1+ carcinomas, as well as between 2+ NA, DE, and 2+ WA carcinomas. When analyzing carcinomas stratified only by the presence of activating mutations in the PI3K-AKT pathway, regardless of the tumor HER2 status, we found that the presence of such a mutation was associated with a lower *ERBB2* expression (*p* = 0.0002).

#### 3.3.2. Gene Expression of the PI3K-AKT, JAK-STAT, and the MAPK Pathway

In the PI3K-AKT pathway analysis (224 genes), H2L carcinomas clustered into two main groups (Figure 2A). One cluster was similar to the profile of HER2-negative carcinomas, which was homogeneous. The other cluster was close to some HER2-positive carcinomas. HER2 0+ and 1+ carcinomas had similar transcriptomic profiles and seemed to express genes in common. DE carcinomas showed a comparable profile to that of 2+ WA carcinomas. HER2 3+ carcinomas were clearly distinguished from the other groups, with a distinct pattern of gene expression unique to this tumor type. HER2 2+ WA carcinomas appeared to be quite similar—but not identical—to HER2 3+ carcinomas.

The components of the JAK-STAT pathway (46 genes) were heterogeneously expressed in H2L carcinomas (Figure 2B). Some of the profiles resembled those of the HER2-positive carcinomas, without, however, a single profile that would be identical. Some others were more similar to some HER2-negative carcinomas, which themselves had heterogeneous profiles, without any cluster. DE carcinomas appeared to have a similar profile to that of 2+ WA carcinomas. HER2 3+ carcinomas had a profile which was closer to that of DE and 2+ WA carcinomas than to those of the other carcinomas, forming a common spectrum. Unlike for the PI3K-AKT pathway, HER2 0+ and 1+ carcinomas had more heterogeneous profiles, without a clear cluster.

Regarding the MAPK pathway (190 genes), H2L carcinomas had heterogeneous profiles, with different tumor clusters (Figure 2C). HER2-negative carcinomas showed more diverse transcriptomic profiles, without a single distinctive cluster, contrary to HER2-positive carcinomas which had a specific and homogeneous transcriptomic profile, clearly distinguishing these carcinomas from other carcinomas. Overall, H2L carcinomas were characterized by the presence of two distinct transcriptomic clusters, one corresponding to the HER2 1+, and the other to 2 + NA and DE carcinomas. DE and 2+ WA carcinomas had similar profiles, with some gene expression similarities to HER2 3+ carcinomas. Carcinomas of the latter subtype, however, formed a clear cluster, showing a homogeneous transcriptomic profile. HER2 1+ and 0+ carcinomas showed more diverse transcriptomic profiles, without any gene expression clustering.

Then, we analyzed expression profiles for each of the three pathways according to the presence of an activating *PIK3CA* mutation, but independently of the tumor HER2 status. In the PI3K-AKT pathway analysis, we found a cluster of mutant carcinomas, mainly composed of 0+, 1+, 2+ NA, and DE carcinomas (Figure 2A). Among H2L and HER2-negative carcinomas, two expression profiles were found depending on the presence or absence of an activating mutation. Mutant H2L carcinomas had a similar profile to that of HER2-negative carcinomas. Non-mutant H2L and HER2-negative carcinomas tended to be similar to HER2-positive carcinomas. The MAPK pathway analysis yielded the same results (Figure 2C). As expected, the results for the JAK-STAT pathway showed much less discriminating profiles, with no separate clusters for mutant and non-mutant carcinomas (Figure 2B).

In the whole transcriptome analysis, we found that DE breast carcinomas had a transcriptomic profile close to that of 2+ WA carcinomas. The profile of H2L carcinomas was comparable to that of HER2-negative carcinomas, with a common gene expression cluster (Appendix A). HER2-positive carcinomas differed from the other two groups, with a specific profile and gene expression unique to these carcinomas.

### 3.4. Global Gene Expression and Phenotypic Profiles within the Different Pathways, and the Impact of PIK3CA Activating Mutations

A gene set enrichment analysis (ssGSEA) was performed to assess the concerted behavior of several defined genes and to detect small changes in gene expression that may explain different phenotypes within the same biological pathways. The expression of each gene in a pathway was measured and assigned a score reflecting its importance in the pathway. The expression of all pathway genes was analyzed to determine their influence on the pathway activation, for each HER2 expression group.

There was a significant difference in gene expression within the three pathways between HER2-negative and HER2-positive carcinomas. We found a stronger activation of all the three pathways in HER2-negative carcinomas, with a decreasing activation gradient from HER2-negative to HER2-positive carcinomas (Figure 3A–D). H2L carcinomas significantly differed from HER2-positive carcinomas regarding the activation of the MAPK pathway (*p* = 0.0073; Figure 3C), with a stronger pathway activation correlating with the activation of various genes in this pathway. No significant difference appeared between HER2-negative and H2L carcinomas; similar overall gene expression profiles resulted in similar levels of activation of the three pathways.

A detailed analysis of the six groups showed similar profiles between HER2 0+ and 1+ carcinomas (expression of the same genes in the same pathways). HER2 3+ carcinomas had a lower activation of the PI3K-AKT pathway than the other groups (*p* < 0.05, Figure 3A), which was consistent with the expression of different genes in this tumor group. Within each of the three pathways, DE carcinomas had a gene enrichment score close to that of 2+ WA carcinomas (many expressed genes in common), with similar activation levels for all the three pathways.

The global activation analysis of the three pathways (ssGSEA) according to the presence of a *PIK3CA*-activating mutation showed a strong activation of signaling pathways in carcinomas carrying such a mutation (Figure 3A–D). The overall activation in the MAPK pathway was significantly higher in carcinomas carrying an activating mutation (*p* = 0.0087; Figure 3C), but not significant in case of a mutation in the PI3K-AKT pathway. However, there was a trend for a higher expression of the PI3K-AKT pathway components by carcinomas with activating mutations, regardless of their HER2 status.

## 4. Discussion

Assessing the HER2 status (expression levels and/or gene amplification) is central to the therapeutic management of breast carcinoma, determining the prescription choices for both conventional and emerging treatments. The current therapeutic landscape considers three main HER2 categories of breast carcinomas: HER2-negative, H2L, and HER2-positive carcinomas. Among those, H2L carcinomas—and especially DE carcinomas—pose therapeutic challenges because they do not seem to respond to conventional anti-HER2 therapies. While new ADCs targeting HER2 (such as T-Dxd) have offered hope in preclinical settings, their effectiveness in patients with H2L carcinomas varies and establishing reliable predictive biomarkers is urgently needed. We aimed at closing this gap by providing an in-depth characterization of H2L carcinomas at a clinical, histologic, and molecular level.

With regard to clinicopathological data, our data confirm the currently known characteristics of H2L carcinomas. We found a trend for bigger tumor sizes with increasing HER2 amplification, without, however, statistical significance, likely due to the small sample size. Still, this trend is in line with studies on larger cohorts which showed that H2L carcinomas were significantly larger than HER2-negative carcinomas [4,36,37]. The histological grades of H2L carcinomas were intermediate between those of HER2-negative and those of HER2-positive carcinomas, as previously reported by others [4,36,37,38,39], in line with significantly lower mitotic and Ki67 proliferation indexes compared with HER2-positive [40]. On the other hand, we found no significant difference in lymph node status, as did Xu and collaborators [40]. Other literature data regarding Ki67 proliferation index [4,36,38,41,42,43] and lymph node status [4,36,37,44,45,46,47] in H2L carcinomas are conflicting. However, these two indexes may depend on factors other than HER2 status, such as the HR status. Indeed, most studies have included HR-negative H2L carcinomas who have a poorer overall prognosis, whereas we only included HR-positive H2L carcinomas (due to the fact that most H2L carcinomas are HR+ [4,36,37,38,39,48,49], and all our DE carcinomas were HR+), so this discordance may be explained by the fact that we only included HR+ carcinomas in our study. Moreover, the HR status may impact response to anti-HER2 treatment independently of the HER2 tumor status [12,13].

Despite numerous studies on H2L breast carcinomas, the biology of these carcinomas is still poorly understood. It remains unclear whether they have a distinct biological profile, different from that of HER2-negative carcinomas, and whether the H2L diagnosis alone may be an independent (adverse or favorable) prognostic factor.

We found a significantly higher *PIK3CA* mutation rate in in H2L than in HER2-positive carcinomas. This is consistent with luminal molecular profiles since *PIK3CA* mutations have been reported to be present in approximately 25–45% of luminal A and B carcinomas, with a much lower prevalence (9%) in basal-like carcinomas [50]. Others demonstrated a differential distribution of certain genes between H2L and HER2-negative carcinomas, with a significant upregulation of luminal genes associated with a downregulation of basal-like proliferation genes in H2L carcinomas compared to HER2-negative carcinomas [4], maintained in HR+ carcinomas. Agostinetto and collaborators [49] found no significant difference between H2L and HER2-negative HR+ carcinomas, in line with our hypothesis that HR+ HER2-negative and HR+ H2L carcinomas are biologically similar.

*PIK3CA* mutations in breast carcinoma were reported to be a favorable prognostic factor, associated with a low histologic grade [51,52,53,54,55], advanced age at diagnosis [52,54], and the absence of lymph node invasion [52,54], with small tumor size [51,52,56], and with a low proliferation index (Ki67) [51,57]. However, few studies specifically addressing *PIK3CA* mutations in H2L carcinomas exist. Our results for H2L carcinomas are consistent with these reports, showing a rather low-grade clinicopathological profile of *PIK3CA*-mutant H2L carcinomas, similar to that of HER2-negative carcinomas. We also found similar *PIK3CA* mutation rates between these two tumor groups in our study, consistent with results previously reported by Denkert and collaborators [42].

Contrary to results reported by others [42], we found only three *TP53* mutations in H2L carcinomas and none in HER2-negative carcinomas. As expected, *TP53*-mutant carcinomas had more aggressive profiles than all H2L carcinomas taken together (histologic grades II and III, and higher Ki67 proliferation indexes). The low prevalence of *TP53* mutations could partly explain the better survival of patients with H2L carcinomas, supporting the hypothesis that H2L and HER2-negative carcinomas have very similar profiles. We found no significant differences in the prevalence of mutations in two other genes of the PI3K-AKT pathway which were previously reported to be linked with a more aggressive tumor behavior, resistance to treatments, and poor prognosis [56,58,59,60,61,62,63]: *AKT1* and *PTEN*, explained by a relatively small sample size in our study.

We highlighted a high correlation between HER2 amplification and elevated *ERBB2* mRNA expression levels, which had also been shown in other studies [4,49], with *ERBB2* expression levels increasing from HER2 IHC group 0+ to 3+. However, we found that the expression rates differed between the different groups of the new HER2 classification. DE carcinomas had *ERBB2* expression levels which were very close to those of 2+ WA carcinomas, suggesting that transcriptomic expression may correlate with 2+ HER2 protein staining with IHC. In contrast, the two subcategories of HER2-positive carcinomas: HER2 2+ WA and 3+ carcinomas, as well as the subgroups of H2L carcinomas HER2 1+, 2+NA, and DE carcinomas, had very different expression levels. Overall, these results did not show a significant difference in the *ERBB2* expression levels between H2L and HER2-negative carcinomas. However, this can be due to a small number of patients included in our study. Indeed, Schettini and collaborators [4] found higher levels of *ERBB2* mRNA in H2L than in HER2-negative HR+ carcinomas, with higher levels in 2+ H2L carcinomas than in 1+ H2L carcinomas. Others found significantly higher expression levels in H2L carcinomas than in HER2-negative carcinomas, with increasing levels from 1+ to 2+ carcinomas [43,49]. Finally, we found a decreased *ERBB2* expression in carcinomas with an activating *PIK3CA* mutation. This may be explained by a very high proportion of HER2-negative and H2L carcinomas compared to HER2-positive carcinomas among mutant carcinomas.

To the best of our knowledge, our study is the first ever reported in the literature to have investigated the signal transduction pathways downstream of the HER2 receptor in H2L carcinomas. We found that H2L carcinomas had transcriptomic profiles which were intermediate between those of HER2-negative and those of HER2-positive carcinomas. It is noteworthy that H2L carcinomas clustered into two different populations: (1) DE carcinomas with profiles that seemed to be closer to those of 2+ NA and 2+ WA carcinomas, and (2) HER2 1+ carcinomas which tended to be closer to 0+ carcinomas. This highlights the diversity of H2L carcinomas’ biological profiles and the absence of a unique transcriptomic cluster specific to the entire H2L category. The fact that HER2-positive carcinomas were clearly distinguishable from H2L and HER2-negative carcinomas is due to the inclusion of HER2 3+ carcinomas which had a transcriptomic behavior of their own, expressing different genes than the other groups, including 2+ WA carcinomas.

Moreover, we found that the degree of activation of the different pathways was inversely correlated with the HER2 amplification. We distinguished three transcriptomic groups: (1) HER2 0+ and 1+ carcinomas; (2) HER2 2+ NA, DE, and 2+ WA carcinomas; and (3) HER2 3+ carcinomas. The transcriptomic profiles thus seem to follow the classical IHC classification and not the new HER2 classification. This suggests that the activation of the three signaling pathways does not depend on the HER2 expression alone, but also on other mechanisms, in particular those related to the presence of activating mutations in genes encoding other components of these pathways, acting downstream of HER2. Consistently, when we focused only on activating mutations in the PI3K-AKT pathway, regardless of the HER2 status, we found that the activation of biological pathways in mutant carcinomas was stronger and seemed to be more efficient than the activation due to HER2 amplification/hyperexpression alone. In the same line, the three transcriptomic groups we found had different mutation rates. HER2 0+ and 1+ carcinomas had a high rate, favoring a strong activation of the PI3K-AKT signaling pathway in the absence of HER2 amplification. In contrast, HER2 3+ carcinomas had a low mutation prevalence and phenotypic profiles related to different gene expression profiles than those of the other groups, secondary to HER2 amplification which activates the different signaling pathways less strongly than activating mutations in the PI3K-AKT pathway. The phenotypic differences between tumor cells can therefore be explained by differences in the number of HER2 receptors on the cell surface and by their mutational status. H2L carcinomas showed heterogeneous gene expression profiles of the PI3K-AKT pathway, according to the presence of an activating mutation: mutant H2L carcinomas had similar profiles to those of HER2-negative carcinomas, while non-mutant H2L carcinomas had similar profiles to those of HER2-positive carcinomas. Taking all this into account, we believe that the pathways’ activation strongly depends on the *PIK3CA* mutational status.

Transcriptomic analysis of the JAK-STAT pathway components did not reveal such clear differences between tumor groups. Indeed, the effect of *PIK3CA* mutations on the JAK-STAT pathway was less strong, consistent with a lower number of associated genes than for the other two pathways. This may explain why the activation of this pathway is less heterogeneous within the groups, with no significant differences between the mutant and non-mutant carcinomas. Therefore, we may conclude that-from a biological point of view-the JAK-STAT pathway activation is not impacted by the presence of a *PIK3CA* mutation, but only by the HER2 amplification status.

Our results have many therapeutic implications. While antibody-dependent cellular cytotoxicity mechanisms partly explain the effect of the conventional anti-HER2 therapy [10], these therapies were not sufficiently effective in H2L carcinomas. More recently, the introduced topoisomerase I inhibitors require a low expression of HER2 on the cell surface. Therefore, they appear to be effective in patients with H2L carcinomas due to HER2 supra-physiological expression, independently of the presence of a *PIK3CA* mutation. However, the fact that we found elevated *PIK3CA* mutation rates in H2L carcinomas raises a question whether PI3K inhibitors could also be effective. Indeed, a recent review highlighted an improved progression-free survival of patients with *PIK3CA*-mutant carcinomas receiving hormonal therapy combined with a PI3K-targeted tyrosine kinase inhibitor [64,65]. Several other studies have focused on the development of pan-PI3K inhibitors (inhibiting all four PI3K class I isoforms: α to δ), α-isoform specific inhibitors (anti-α PI3K: alpelisib, taselisib), or AKT inhibitors (reviewed in [66,67,68]). Still, given that a majority of H2L carcinomas are HR-positive, the development of hormone resistance due to the above-mentioned mutations could be enhanced. Moreover, *PIK3CA* mutations, as well as the underlying signaling pathway involving AKT1 and mTOR, are linked to resistance to chemotherapy, to hormone therapy [56,58,59,60], and to anti-HER2 targeted therapies [61,62,63]. It would be interesting to determine the overall prevalence of *PIK3CA*-activating mutations in H2L carcinomas as well as to study the efficacy of treatments inhibiting this pathway.

Our study also has a number of limitations. It is a monocentric retrospective study with all the known biases inherent in this design. In addition, we chose to include only HR+ H2L carcinomas on the grounds of frequency. Our results cannot be transposed to H2L HR- carcinomas. Survival and treatment data could not be obtained in this study, notably due to the absence of sufficiently long follow-up and a lack of data concerning responses to conventional anti-HER2 treatments, ADCs, and hormone therapy.

Further studies on a larger cohort seem necessary to confirm these preliminary data.

## 5. Conclusions

Even though the retrospective monocentric design with the associated potential biases is a limitation of our study, it is—to our knowledge—the first study comparing molecular and transcriptomic profiles of H2L breast carcinomas with those of HER2-negative and HER2-positive carcinomas, and analyzing the main intracellular signaling pathways linked to HER2 expression. We found that H2L HR+ carcinomas present histopathological and molecular characteristics close to those of HER2-negative HR+ carcinomas. Transcriptomic profiles of these two carcinoma groups are similar and depend largely on the presence of an activating mutation in the PI3K-AKT pathway, causing—in a majority of cases—a strong activation of three intracellular signaling pathways: PI3K-AKT, JAK-STAT, and MAPK. The presence of an activating mutation downstream of the HER2 receptor appears to be more important for determining tumor behavior than the HER2 amplification status. Since it is known that the presence of a *PIK3CA* mutation causes resistance to hormonal therapy and to classical anti-HER2 therapies, these results could help explain differences in the effectiveness of conventional and innovative treatments, including new antibody-drugs conjugates (T-DXd) in patients with *PIK3CA*-mutant HER2-negative and H2L carcinomas. They also raise a question whether including PI3K-AKT pathway inhibitors in the therapeutic management of patients with these H2L HR+ carcinomas would not be beneficial for these patients. Further studies on larger cohorts of patients are needed to validate these hypotheses.

## Figures and Tables

**Figure 1 cancers-16-02009-f001:**
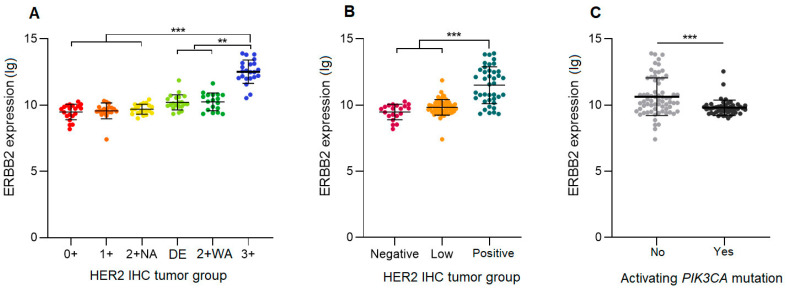
*ERBB2* mRNA expression levels in hormone-receptor-positive breast carcinomas. (**A**) HER2 double equivocal (DE) carcinomas compared to five other HER2 immunohistochemistry ASCO/CAP groups. (**B**) H2L carcinomas compared to HER2-negative and HER2-positive carcinomas. (**C**) Carcinomas stratified by the presence or absence of a *PIK3CA* activating mutation. 2 + NA: HER2 non-amplified; 2 + WA: HER2 weakly amplified; **: *p* < 0.005; ***: *p* < 0.0001.

**Figure 2 cancers-16-02009-f002:**
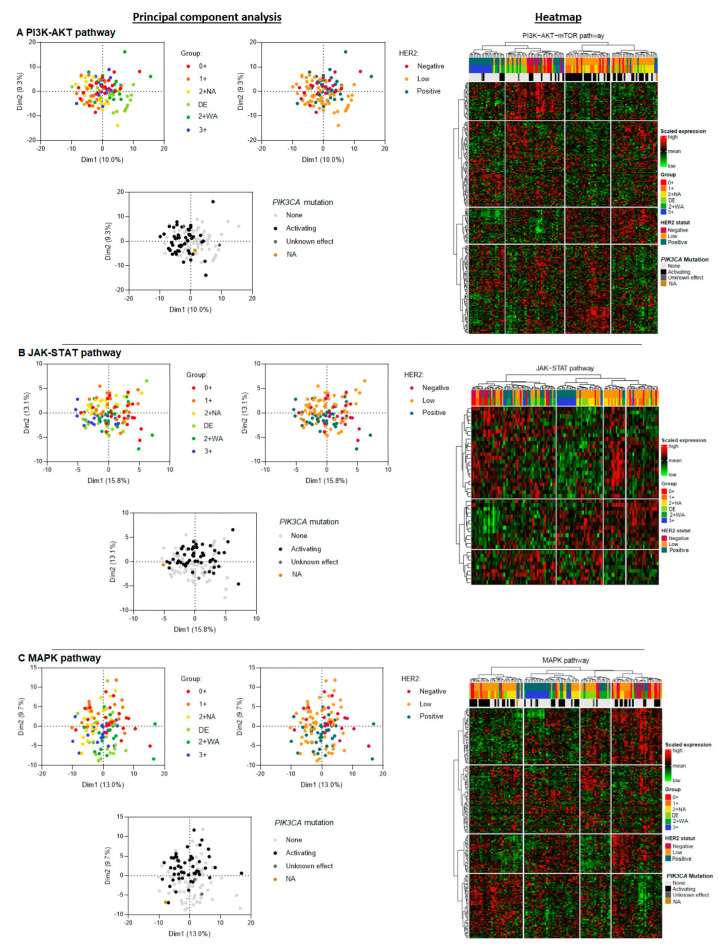
Principal component analysis of three signaling pathways downstream of HER2: PI3K-AKT (**A**), JAK-STAT (**B**), and MAPK (**C**) in hormone-receptor-positive breast carcinomas—a comparison of three HER2 carcinoma groups (H2L, HER2-negative, and HER2-positive), then of six ASCO/CAP HER2 groups, and finally of carcinomas stratified by the presence of an activating *PIK3CA* mutation. DE: HER2 double-equivocal carcinomas; 2+ NA: non-amplified carcinomas; 2+ WA: weakly amplified carcinomas. NA: data not available.

**Figure 3 cancers-16-02009-f003:**
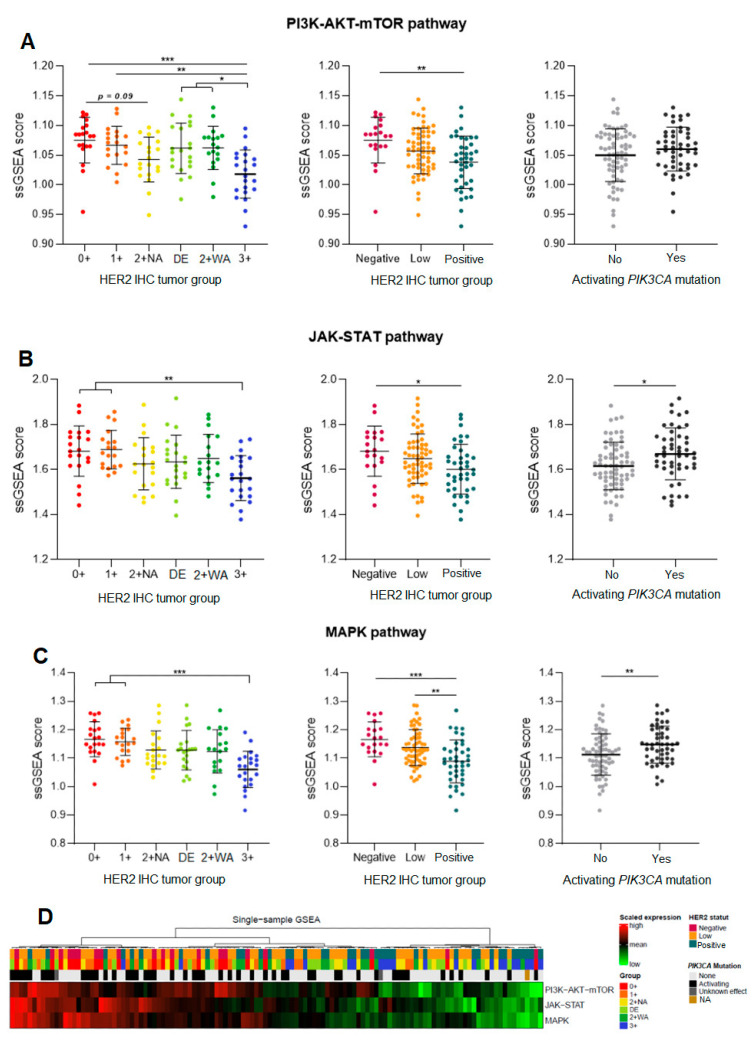
Single-sample Gene Set Enrichment Analysis (ssGSEA) of three signaling pathways downstream of HER2: PI3K-AKT (**A**), JAK-STAT (**B**), and MAPK (**C**), in hormone-receptor-positive breast carcinomas—a comparison of three HER2 tumor groups (H2L, HER2-negative, and HER2-positive), then of six ASCO/CAP HER2 groups, and finally of carcinomas stratified by the presence of an activating *PIK3CA* mutation. (**D**) Heatmap for the three pathways. DE: HER2 double-equivocal carcinomas; 2 + NA: non-amplified carcinomas; 2 + WA: weakly amplified carcinomas. NA: data not available. * *p* < 0.05; ** *p* < 0.005; *** *p* < 0.0005.

**Table 1 cancers-16-02009-t001:** Clinicopathological characteristics of H2L breast carcinoma patients and carcinomas compared to HER2-negative and HER2-positive groups.

				*p*-Values			
	H2L (*n* = 62)	HER2-Negative (*n* = 20)	HER2-Positive (*n* = 43)	All	0+ vs. HER2-Positive	0+ vs. H2L	HER2-Positive vs. H2L
	*n* (%)	*n* (%)	*n* (%)				
Patient age				0.9054			
Mean ± SD	65.0 ± 12.6	65.3 ± 12.8	64.0 ± 13.5				
Median [min–max]	67.0 [34.0–93.0]	68.5 [36.0–91.0]	64.0 [37.0–89.0]				
Patient menopausal status				0.4422			
Peri/premenopausal	10 (16.1)	5 (25.0)	11 (25.6)				
Postmenopausal	52 (83.9)	15 (75.0)	32 (74.4)				
Tumor size (US, mm)				0.126			
Mean ± SD	18.9 ± 14.1	14.9 ± 12.3	18.7 ± 11.4				
Median [min–max]	15.0 [5.0–70.0]	11.3 [3.5–50.0]	15.0 [4.0–58.0]				
Node status (US)				0.5397			
N0	59 (95.2)	18 (90.0)	38 (88.4)				
N1	3 (4.8)	2 (10.0)	4 (9.3)				
N3	0 (0)	0 (0)	1 (2.3)				
Multifocality				0.5519			
Unifocal	50 (80.6)	18 (90.0)	37 (86.0)				
Bifocal	12 (19.4)	2 (10.0)	6 (14.0)				
Tumor size (clinical, mm)				0.8995			
Mean ± SD	18.1 ± 11.9	18.1 ± 11.7	18.9 ± 11.6				
Median [min–max]	15.0 [4.5–70.0]	15.5 [6.0–55.0]	15.0 [5.8–58.0]				
Node status				0.0146	0.0497	0.0531	0.6476
pN0	41 (66.1)	9 (45.0)	34 (79.1)				
pN1	19 (30.6)	6 (30.0)	7 (16.3)				
pN2	2 (3.2)	4 (20.0)	2 (4.7)				
pN3	0 (0)	1 (5.0)	0 (0)				
E&E grade				0.0005	0.0005	0.9656	0.0017
I	25 (40.3)	10 (50.0)	3 (7.0)				
II	31 (50.0)	10 (50.0)	31 (72.1)				
III	6 (9.7)	0 (0)	9 (20.9)				
Glandular differentiation				0.1734			
1	3 (4.8)	1 (5.0)	0 (0)				
2	30 (48.4)	9 (45.0)	14 (32.6)				
3	29 (46.8)	10 (50.0)	29 (67.4)				
Nuclear grade				0.0348	0.1398	1	0.0756
1	2 (3.2)	0 (0)	0 (0)				
2	54 (87.1)	19 (95.0)	31 (72.1)				
3	6 (9.7)	1 (5.0)	12 (27.9)				
Mitosis score				0.0003	0.0001	0.0651	0.0537
1	39 (62.9)	19 (95.0)	15 (34.9)				
2	14 (22.6)	1 (5.0)	18 (41.9)				
3	9 (14.5)	0 (0)	10 (23.3)				
Mitotic index (/mm²)				0.0002	0.0002	0.4007	0.0042
Mean ± SD	2.9 ± 2.6	1.8 ± 1.6	4.8 ± 3.2				
Median [min–max]	1.8 [0.4–10.5]	0.9 [0.4–6.6]	4.4 [0.4–15.1]				
Histologic subtype				0.5616			
Micropapillary	2 (3.2)	0 (0)	1 (2.3)				
Mucinous	0 (0)	1 (5.0)	0 (0)				
NST	57 (91.9)	18 (90.0)	41 (95.3)				
NST + micropapillary	3 (4.8)	1 (5.0)	1 (2.3)				
Lymphovascular emboli				0.9027			
No	41 (66.1)	13 (65.0)	30 (69.8)				
Yes	21 (33.9)	7 (3.0)	13 (30.2)				
sTIL (%)				0.1407			
Mean ± SD	7.0 ± 7.6	8.2 ± 9.3	10.6 ± 9.6				
Median [min–max]	5.0 [1.0–50.0]	4.0 [1.0–30.0]	5.0 [1.0–40.0]				
sTIL (≤10%)				0.0507			
No	8 (12.9)	4 (20.0)	14 (32.6)				
Yes	54 (87.1)	16 (80.0)	29 (67.4)				
sTILs (>40%)				1			
No	61 (98.4)	20 (100.0)	42 (97.7)				
Yes	1 (1.6)	0 (0)	1 (2.3)				
ER (I × %)				0.5524			
Mean ± SD	285.2 ± 34.0	280.0 ± 41.9	277.3 ± 43.2				
Median [min–max]	300.0 [180.0–300.0]	300.0 [160.0–300.0]	300.0 [140.0–300.0]				
PR (I × %)				0.0281	0.0757	0.7856	0.0567
Mean ± SD	198.4 ± 99.4	212.3 ± 100.2	151.0 ± 110.4				
Median [min–max]	210.0 [0.0–300.0]	247.5 [20.0–300.0]	140.0 [0.0–300.0]				
Ki67 (%)				<0.0001	0.0002	0.3919	0.0003
n	62	20	42 *				
Mean ± SD	15.7 ± 10.2	14.4 ± 13.5	23.8 ± 11.4				
Median [min–max]	14.5 [2.0–60.0]	11.0 [2.0–60.0]	21.5 [5.0–60.0]				

ER: estrogen receptor; E&E: Bloom–Richardson histological grade modified by Elston and Ellis; HER2: human epidermal growth factor receptor 2; NST: non special type; PR: progesterone receptor; SD: Standard Deviation; sTILs: stromal tumor infiltrating lymphocytes; US: ultrasound. *** Missing data due to lack of tumor material.

**Table 2 cancers-16-02009-t002:** Clinicopathological characteristics of HER2 double equivocal (DE) breast carcinoma patients and carcinomas compared to the five other ASCO/CAP HER2 groups.

							*p*-Values		
	HER2 DE (*n* = 22)	HER2 0+ (*n* = 20)	HER2 1+ (*n* = 20)	HER2 2+ NA (*n* = 20)	HER2 2+ WA (*n* = 20)	HER2 3+ (*n* = 23)	All	DE vs. 0+	DE vs. 1+
	*n* (%)	*n* (%)	*n* (%)	*n* (%)	*n* (%)	*n* (%)			
Patient age							0.1129		
Mean ± SD	69.5 ± 9.1	65.3 ± 12.8	66.7 ± 11.5	58.3 ± 14.6	63.3 ± 13.3	64.7 ± 13.9			
Median [min–max]	70.5 [50.0–85.0]	68.5 [36.0–91.0]	71.0 [42.0–83.0]	53.0 [34.0–93.0]	62.0 [37.0–84.0]	64.0 [38.0–89.0]			
Patient menopausal status							0.5713		
Peri/premenopausal	2 (9.1%)	5 (25.0%)	3 (15.0%)	5 (25.0%)	6 (30.0%)	5 (21.7%)			
Postmenopausal	20 (90.9%)	15 (75.0%)	17 (85.0%)	15 (75.0%)	14 (70.0%)	18 (78.3%)			
Tumor size (US, mm)							0.0533		
Mean ± SD	20.2 ± 14.1	14.9 ± 12.3	13.4 ± 8.4	22.9 ± 17.3	16.6 ± 8.3	20.5 ± 13.4			
Median [min–max]	15.0 [6.0–70.0]	11.3 [3.5–50.0]	11.0 [5.0–38.0]	16.5 [6.0–70.0]	14.5 [4.0–30.0]	15.0 [7.0–58.0]			
Node status (US)							0.3889		
N0	19 (86.4%)	18 (90.0%)	20 (100.0%)	20 (100.0%)	18 (90.0%)	20 (87.0%)			
N1	3 (13.6%)	2 (10.0%)	0 (0.0%)	0 (0.0%)	2 (10.0%)	2 (8.7%)			
N3	0 (0.0%)	0 (0.0%)	0 (0.0%)	0 (0.0%)	0 (0.0%)	1 (4.3%)			
Multifocality							0.1893		
Unifocal	18 (81.8%)	18 (90.0%)	14 (70.0%)	18 (90.0%)	15 (75.0%)	22 (95.7%)			
Bifocal	4 (18.2%)	2 (10.0%)	6 (30.0%)	2 (10.0%)	5 (25.0%)	1 (4.3%)			
Tumor size (clinical, mm)							0.6973		
Mean ± SD	17.5 ± 6.7	18.1 ± 11.7	14.6 ± 7.3	22.1 ± 17.9	17.8 ± 8.7	19.9 ± 13.7			
Median [min–max]	16.0 [8.0–30.0]	15.5 [6.0–55.0]	12.3 [4.5–32.0]	17.2 [5.0–70.0]	14.8 [7.0–35.0]	15.0 [5.8–58.0]			
Node status							0.1845		
pN0	12 (54.5%)	9 (45.0%)	14 (70.0%)	15 (75.0%)	15 (75.0%)	19 (82.6%)			
pN1	9 (40.9%)	6 (30.0%)	6 (30.0%)	4 (20.0%)	4 (20.0%)	3 (13.0%)			
pN2	1 (4.5%)	4 (20.0%)	0 (0.0%)	1 (5.0%)	1 (5.0%)	1 (4.3%)			
pN3	0 (0.0%)	1 (5.0%)	0 (0.0%)	0 (0.0%)	0 (0.0%)	0 (0.0%)			
E&E grade							<0.0001	0.2706	0.0577
I	3 (13.6%)	10 (50.0%)	12 (60.0%)	10 (50.0%)	2 (10.0%)	1 (4.3%)			
II	17 (77.3%)	10 (50.0%)	7 (35.0%)	7 (35.0%)	14 (70.0%)	17 (73.9%)			
III	2 (9.1%)	0 (0.0%)	1 (5.0%)	3 (15.0%)	4 (20.0%)	5 (21.7%)			
Glandular differentiation							0.0166	1	0.398
1	1 (4.5%)	1 (5.0%)	0 (0.0%)	2 (10.0%)	0 (0.0%)	0 (0.0%)			
2	5 (22.7%)	9 (45.0%)	12 (60.0%)	13 (65.0%)	7 (35.0%)	7 (30.4%)			
3	16 (72.7%)	10 (50.0%)	8 (40.0%)	5 (25.0%)	13 (65.0%)	16 (69.6%)			
Nuclear grade							0.0349	1	1
1	0 (0.0%)	0 (0.0%)	1 (5.0%)	1 (5.0%)	0 (0.0%)	0 (0.0%)			
2	20 (90.9%)	19 (95.0%)	18 (90.0%)	16 (80.0%)	17 (85.0%)	14 (60.9%)			
3	2 (9.1%)	1 (5.0%)	1 (5.0%)	3 (15.0%)	3 (15.0%)	9 (39.1%)			
Mitosis score							<0.0001	0.0156	0.2428
1	10 (45.5%)	19 (95.0%)	17 (85.0%)	12 (60.0%)	6 (30.0%)	9 (39.1%)			
2	9 (40.9%)	1 (5.0%)	3 (15.0%)	2 (10.0%)	9 (45.0%)	9 (39.1%)			
3	3 (13.6%)	0 (0.0%)	0 (0.0%)	6 (30.0%)	5 (25.0%)	5 (21.7%)			
Mitotic index (/mm²)							<0.0001	0.0065	0.1072
Mean ± SD	4.2 ± 2.6	1.8 ± 1.6	2.4 ± 2.7	2.0 ± 2.1	4.9 ± 3.3	4.6 ± 3.3			
Median [min–max]	4.1 [0.9–10.5]	0.9 [0.4–6.6]	1.3 [0.4–9.6]	0.7 [0.4–7.1]	5.0 [0.4–11.3]	4.0 [0.5–15.1]			
Histologic subtype							0.0988		
Micropapillary	2 (9.1%)	0 (0.0%)	0 (0.0%)	0 (0.0%)	1 (5.0%)	0 (0.0%)			
Mucinous	0 (0.0%)	1 (5.0%)	0 (0.0%)	0 (0.0%)	0 (0.0%)	0 (0.0%)			
NST	17 (77.3%)	18 (90.0%)	20 (100.0%)	20 (100.0%)	19 (95.0%)	22 (95.7%)			
NST + micropapillary	3 (13.6%)	1 (5.0%)	0 (0.0%)	0 (0.0%)	0 (0.0%)	1 (4.3%)			
Lymphovascular emboli							0.1758		
No	11 (50.0%)	13 (65.0%)	17 (85.0%)	13 (65.0%)	12 (60.0%)	18 (78.3%)			
Yes	11 (50.0%)	7 (35.0%)	3 (15.0%)	7 (35.0%)	8 (40.0%)	5 (21.7%)			
sTIL (%)							0.0429	1	0.6094
Mean ± SD	6.4 ± 5.8	8.2 ± 9.3	10.0 ± 10.6	4.7 ± 4.7	10.2 ± 11.2	11.0 ± 8.3			
Median [min–max]	5.0 [1.0–20.0]	4.0 [1.0–30.0]	6.5 [1.0–50.0]	2.0 [1.0–20.0]	5.0 [1.0–40.0]	10.0 [2.0–25.0]			
sTIL (≤10%)							0.1314		
No	3 (13.6%)	4 (20.0%)	4 (20.0%)	1 (5.0%)	5 (25.0%)	9 (39.1%)			
Yes	19 (86.4%)	16 (80.0%)	16 (80.0%)	19 (95.0%)	15 (75.0%)	14 (60.9%)			
sTILs (>40%)							0.4702		
No	22 (100.0%)	20 (100.0%)	19 (95.0%)	20 (100.0%)	19 (95.0%)	23 (100.0%)			
Yes	0 (0.0%)	0 (0.0%)	1 (5.0%)	0 (0.0%)	1 (5.0%)	0 (0.0%)			
ER (I × %)							0.0299	0.9142	1
Mean ± SD	292.7 ± 22.5	280.0 ± 41.9	292.0 ± 23.5	270.0 ± 47.4	290.0 ± 30.8	266.3 ± 49.8			
Median [min–max]	300.0 [200.0–300.0]	300.0 [160.0–300.0]	300.0 [200.0–300.0]	300.0 [180.0–300.0]	300.0 [200.0–300.0]	300.0 [140.0–300.0]			
PR (I × %)							0.0407	0.833	1
Mean ± SD	180.5 ± 102.5	212.3 ± 100.2	184.0 ± 95.5	232.5 ± 95.7	133.1 ± 119.6	166.5 ± 101.8			
Median [min–max]	170.0 [0.0–300.0]	247.5 [20.0–300.0]	160.0 [30.0–300.0]	270.0 [10.0–300.0]	130.0 [0.0–300.0]	160.0 [20.0–300.0]			
Ki67 (%)							<0.0001	0.0175	0.0011
n	22	20	20	20	20	22 *			
Mean ± SD	20.8 ± 10.4	14.4 ± 13.5	10.6 ± 5.7	15.2 ± 11.1	20.0 ± 7.8	27.2 ± 13.1			
Median [min–max]	20.0 [8.0–60.0]	11.0 [2.0–60.0]	9.0 [5.0–25.0]	13.5 [2.0–40.0]	20.0 [5.0–40.0]	25.0 [8.0–60.0]			

2+ NA: HER2 non-amplified; 2+ WA: weakly amplified; ER: estrogen receptor; E&E: Bloom–Richardson histological grade modified by Elston and Ellis; HER2: human epidermal growth factor receptor 2; NST: non special type; PR: progesterone receptor; SD: Standard Deviation; sTILs: stromal tumor infiltrating lymphocytes; US: ultrasound. *** Missing data due to lack of tumor material.

**Table 3 cancers-16-02009-t003:** Mutation rates in key breast cancer genes in H2L breast carcinomas compared to HER2-negative and HER2-positive carcinomas.

			Controls		*p*-Value	
Gene	Mutation Impact *	H2L (n = 62)	HER2-Negative (n = 20)	HER2-Positive (n = 40)	All	H2L vs. HER2-Positive
		n (%)	n (%)	n (%)		
*PIK3CA*	Gain of function	28 (45.2)	8 (40.0)	6 (15.0)	**0.0063**	**0.0048**
*AKT1*	Gain of function	5 (8.1)	0	0	0.1104	-
*PTEN*	Loss of function	1 (1.6)	0	1 (2.5)	1	-
*TP53*	Loss of function	3 (4.8)	0	12 (30.0)	**0.0003**	**0.0028**
*BRCA1*	-	0	0	0	-	-
*BRCA2*	Loss of function	3 (4.8)	0	2 (5.0)	0.854	-
*PALB2*	-	0	0	0	-	-
*ARID1A*	Loss of function	0	0	1 (2.5)	0.4918	-
*KRAS*	-	0	0	0	-	-
*NRAS*	-	0	0	0	-	-
*BRAF*	-	0	0	0	-	-

HER2: human epidermal growth factor receptor 2; * impact defined based on literature data; in bold: *p*-values indicative of a statistical significance.

**Table 4 cancers-16-02009-t004:** Mutation rates in key breast cancer genes in HER2 double-equivocal (DE) carcinomas compared to the five other ASCO/CAP HER2 groups.

			Controls					
Gene	Mutation Impact *	HER2 DE (n = 22)	HER2 0+ (n = 20)	HER2 1+ (n = 20)	HER2 2+ NA (n = 20)	HER2 2+ WA (n = 20)	HER2 3+ (n = 20)	*p*-Value
		n (%)	n (%)	n (%)	n (%)	n (%)	n (%)	
*PIK3CA*	Gain of function	8 (36.4)	8 (40.0)	10 (50.0)	10 (50.0)	5 (25.0)	1 (5.0)	**0.0227**
*AKT1*	Gain of function	0	0	2 (10.0)	3 (15)	0	0	**0.0368**
*PTEN*	Loss of function	0	0	0	1 (5.0)	1 (5.0)	0	0.7019
*TP53*	Loss of function	1 (4.5)	0	0	2 (10.0)	4 (20.0)	8 (40.0)	**0.0004**
*BRCA1*	-	0	0	0	0	0	0	
*BRCA2*	Loss of function	2 (9.1)	0	0	1 (5.0)	1 (5.0)	1 (5.0)	0.8997
*PALB2*	-	0	0	0	0	0	0	
*ARID1A*	Loss of function	0	0	0	0	0	1 (5.0)	0.8197
*KRAS*	-	0	0	0	0	0	0	
*NRAS*	-	0	0	0	0	0	0	
*BRAF*	-	0	0	0	0	0	0	

HER2: human epidermal growth factor receptor 2; 2+ NA: non-amplified; 2+ WA: weakly amplified; * impact defined based on literature data; in bold: *p*-values indicative of a statistical significance.

## Data Availability

All data generated or analyzed during this study are included in this article or in the Appendix A.

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
