# Peer review of "HER2-Low Luminal Breast Carcinoma Is Not a Homogenous Clinicopathological and Molecular Entity"

_cancers, 2024, doi:10.3390/cancers16112009_

Round 1
Reviewer 1 Report
Comments and Suggestions for Authors
Determination of the HER2 status is essential for the therapeutic management of breast cancer patients. Breast carcinomas were according to HER2 expression and gene amplification previously divided in two categories: HER2-negative and HER2-positive. In addition, a category of HER2-low has recently emerged, and is assigned to tumors with IHC assay score of 1+ or 2+/ISH negative, predominantly with positive expression of hormone receptors.
With the introduction of new antibody-drug conjugates (ADCs) for the treatment of HER2-low breast carcinomas characterization of different levels of HER2 expression in breast cancer has become one of the clinical necessities. Although numerous studies on HER-2 low breast carcinomas are currently ongoing, the biology of these tumors is still not completely understood therefore the presented study has major clinical significance. The authors analyzed mutational status and transcriptomic activities of three HER2 effector pathways: PI3K-AKT, MAPK, and JAK-STAT, in association with clinicopathologic features, in HER2-low tumors and compared them to HER2-positive and HER2-negative tumors.
The article is well written, fluent and well organized. It is also very clear and detailed for the most part. The conclusions drawn from the study are well supported by the results presented. The references are up to date.
Major Objection
1. The sample size is comprised of: 62 HER2-low, 43 HER2-positive and 20 HER2-negative tumors. Can the authors explain why they did not include approximately equal number of samples in each group?
Minor Objection
1. Among the criteria for patient exclusion the authors stated that “patients who received neoadjuvant therapy without prior biopsy” were excluded from the study but since neoadjuvant therapy must not be introduced without prior biopsy I suppose that the results of prior biopsy were not available?
Author Response
Thank you very much for taking the time to review this manuscript. Please find the detailed responses below.
Major Objection
- The sample size is comprised of: 62 HER2-low, 43 HER2-positive and 20 HER2-negative tumors. Can the authors explain why they did not include approximately equal number of samples in each group?
Thank you for pointing this out. The initial idea of the study was to characterize double-equivocal HER2 carcinomas. Immediately after inclusion of the cases, we decided to focus on the characterization of HER2-low carcinomas, which seemed more relevant in the current clinical context.
As a result, 22 double-equivocal HER2 carcinomas were initially included (corresponding to all carcinomas in this group diagnosed between January 2018 and December 2020 at the Georges-François Leclerc Cancer Center France), and then 20 carcinomas from each of the other groups were added for analysis: 20 HER2 0+, 20 HER2 1+, 20 HER2 2+ non-amplified, 20 HER2+ amplified and 23 HER2 3+. The different groups were then combined to obtain the HER2-low and HER2-positive categories.
Minor Objection
- Among the criteria for patient exclusion the authors stated that “patients who received neoadjuvant therapy without prior biopsy” were excluded from the study but since neoadjuvant therapy must not be introduced without prior biopsy I suppose that the results of prior biopsy were not available?
Exactly, these were patients for whom the prior biopsy was not available (biopsy from another center without the possibility of recovering the paraffin block, or lack of material of the paraffin block not allowing the study's molecular techniques to be performed). This clarification has been added to the manuscript (line 144).
You will find attached the new version of the manuscript, taking into account the comments of all the reviewers.

Reviewer 2 Report
Comments and Suggestions for Authors
General comments:
The authors of the manuscript have comprehensively described the profiles of all subtypes of HR-positive breast cancer, with a particular focus on HER2-low breast cancer. It is already known that the newest entity in this field, HER2 low breast cancer, is not a uniform disease and therefore the study that the authors undertook was useful. They looked closely at the characteristics of this tumor subtype and compared them with other HR-positive breast cancers, regardless of HER2 status. They found some clinicopathological differences and also differences in molecular profiles, such as mutational status and transcriptomic profiles. Regarding the clinicopathological differences of HER2 low tumors, they found nothing new, all their findings were already known (such as higher tumor grade, higher mitotic index .... in HER2 positive tumors). However, the exact molecular profile of these tumors was less known and could explain the diversity of this tumor subtype. The relatively high prevalence of the PIK3CA gain-of-function mutation in HER2 low tumors and its potential role in response to some new therapies warrants/calls for further studies on this topic. The second most common mutation found in the whole group of tumors was a P53 mutation, which was most common in the HER2-positive subtype, also confirming an already known fact. The three HER2 effector intracellular signaling pathways have been thoroughly investigated and the results seem interesting. The PIK3-AKT and potentially also the MAPK signaling pathway, the activity of two of them might have a potential role in driving the diversity, but most probably not the JAK-STAT pathway.
The manuscript is long and difficult to read. Therefore, I recommend that it be written in a more concise and comprehensive manner, focusing on the subject of the study and the originally defined objectives: the paper needs to be shortened.
The manuscript needs extensive English editing, either by a native speaker or a person with an English editing certificate.
Nevertheless, a great deal of work has been done and the authors are to be congratulated.
I would like to point out some additional/specific shortcomings.
The manuscript uses three terms to describe the same thing: carcinoma and cancer and tumor. Semantically, they are all correct, but I suggest sticking to a consistent wording. "Clinical-pathological": should be written the same way throughout the manuscript, either with or without a hyphen.
Some abbreviations are not explained, such as NST (Table 2)
The abbreviation H2L is explained the first time it is used, but the abbreviation is not used in some places when it should be (instead of the abbreviation, the non-abbreviated description "HER2 low" is used).
Simple summary:
"With the advent of antibody drugs ...." I suggest adding "some new". Only the latest generation of ADCs showed activity in HER2 low tumors, not the older one (T-DM1).
Introduction:
There are only three references. Many references are missing and some are not used properly, such as reference #2 on line 63. The fact that it is referenced is also written in this article, but the reference cited is not a source reference. The three intracellular pathways are explained in detail in the introduction, making the introduction too long. Anyway, there are no references at all for the written statements.
Line 64: "the majority...."? I guess ALL of them.
Line 80: "anti-HER2 conjugated antibody linked..." Omit the word conjugated.
Line 91: "kinases, conditions three...." Improve the wording, e.g. "influences" or "plays a crucial role".
Materials and Methods:
This section should be more structured (paragraph 2, 3, 4).
In paragraphs 2.1. to 2.5. references should be added where appropriate.
Results:
Table 1 and Table 2: Tumor size (US) and tumor size? US size is clear, but the other value is not clear. Is it clinical size or pathology report? If it is pathology report, the nine neoadjuvant cases cannot be included. The same goes for nodal status. All other reported characteristics are based on surgical specimens or CNBs before systemic treatment. If not, this must be corrected.
Discussion:
The discussion section is long. Try to be more selective in your discussion.
ADCs is abbreviated for the first time in this section, although it appears in the previous sections.
T-DXd is the correct abbreviation for trastuzumab-deruxtecan (not T-DXd). IT is not consistent throughout the manuscript.
Line 480: three references ..... only one reference is written #22
Line 600: "first anti-HER2 treatments" ..... In the introduction section this therapy is called "conventional anti-HER2 therapy". I suggest using the same wording.
The discussion goes into detail about the signaling pathways and their potential role in the clinic. In the last paragraph, when discussing the potential therapeutic implications of the findings, I fully agree with the authors that the findings may partly explain the efficacy/non-efficacy of conventional anti-HER2 therapies in PIK3CA mutant cancers and that the addition of PI3K inhibitors may potentially improve the outcome of these patients. The study the authors refer to is #33. This is one of the studies that looked at the efficacy of a PI3K inhibitor, a phase 2 study in the HER2-negative breast cancer population. But we already have data on the efficacy of alpelisib in combination with fulvestrant (phase 3 trial, although only for PFS, significant) in this (HER2 negative according to the old, dichotomous HER2 classification) patient population. Specific efficacy in the HER2 low population is still in question.
The Discussion section, although too long, lacks the paragraph "limitations/restrictions of the study".
The small sample size, lack of correlation with response to anti-HER2 therapy, conventional and/or ADCs, and endocrine therapy need to be highlighted. Data on outcome in relation to active pathways would provide a more comprehensive insight into the clinical relevance of the findings. This is important because although the new generation of anti-HER2 therapies target the HER2 protein, the by-stander effect they induce plays an important (crucial?) role. Therefore, the present work could be the backbone for further research in this area.
Comments on the Quality of English Language
The manuscript needs extensive English editing, either by a native speaker or a person with an English editing certificate.
Author Response
Thank you very much for taking the time to review this manuscript. Please find the detailed responses below and the corresponding revisions in the re-submitted files.
1) First, for quality on English language, the manuscript has already been translated and fully proofread by a native speaker. Would you like a second review?
2) "The manuscript uses three terms to describe the same thing: carcinoma and cancer and tumor. Semantically, they are all correct, but I suggest sticking to a consistent wording. "Clinical-pathological": should be written the same way throughout the manuscript, either with or without a hyphen. Some abbreviations are not explained, such as NST (Table 2). The abbreviation H2L is explained the first time it is used, but the abbreviation is not used in some places when it should be (instead of the abbreviation, the non-abbreviated description "HER2 low" is used)."
I agree with these comments. Therefore, I have harmonized all the words "carcinoma, cancer and tumor". Similarly, the word "clinical-pathological" has been written in the same way throughout the manuscript. The words “carcinoma” and “clinicopathological” have been chosen to respect the title of the article.
The abbreviation NST has also been added at the bottom of Tables 1 and 2 (NST: non special type).
The term H2L has been added wherever necessary, in accordance with your comments.
All these modifications appear in red in the manuscript.
3) "Simple summary: "With the advent of antibody drugs ...." I suggest adding "some new". Only the latest generation of ADCs showed activity in HER2 low tumors, not the older one (T-DM1)."
Agree, only the latest ADCs have proven their effectiveness. I have, accordingly, changed this sentence to emphasize this point (line 27).
4) "Introduction: There are only three references. Many references are missing and some are not used properly, such as reference #2 on line 63. The fact that it is referenced is also written in this article, but the reference cited is not a source reference. The three intracellular pathways are explained in detail in the introduction, making the introduction too long. Anyway, there are no references at all for the written statements.
Thank you for your comments. For all the references in the article, I had tried to reduce them so as not to include too many. As a result, I've added all the references I think are necessary in the text. As for the reference on line 63, after checking, it is indeed the wrong one: the change has been made.
Indeed, the description of the three signaling pathways adds a lot of text to the introduction. Taking your comment into account, I've tried to reduce this part (lines 92-101).
5) "Line 64: "the majority...."? I guess ALL of them."
I totally agree, all these carcinomas express HER2 on the cell membrane surface to a variable degree (line 64).
6) "Line 80: "anti-HER2 conjugated antibody linked..." Omit the word conjugated."
Very good, the change is made (line 80).
7) "Line 91: "kinases, conditions three...." Improve the wording, e.g. "influences" or "plays a crucial role"."
Very well (line 91).
8) "Materials and Methods: This section should be more structured (paragraph 2, 3, 4). In paragraphs 2.1. to 2.5. references should be added where appropriate."
Regarding the structuring of the section, sub-sections have already been created to distinguish the different parts of the methodology (clinical, histological, immunohistochemical, in situ hybridization and molecular). Could you tell me what you would have liked to see in this part?
References have been added following your comment.
9) "Results: Table 1 and Table 2: Tumor size (US) and tumor size? US size is clear, but the other value is not clear. Is it clinical size or pathology report? If it is pathology report, the nine neoadjuvant cases cannot be included. The same goes for nodal status. All other reported characteristics are based on surgical specimens or CNBs before systemic treatment. If not, this must be corrected."
Indeed, this is not made clear in the text. US size corresponds to ultrasound size, while tumor size (currently unspecified) corresponds to clinical size. I've corrected this in both tables.
10) "Discussion: The discussion section is long. Try to be more selective in your discussion. ADCs is abbreviated for the first time in this section, although it appears in the previous sections. T-DXd is the correct abbreviation for trastuzumab-deruxtecan (not T-DXd). It is not consistent throughout the manuscript."
Concerning the two abbreviations mentioned above, it seems to me that the term ADCs is abbreviated for the first time in the introduction (line 79 and then line 467 and 663). Similarly, I can only find the abbreviation T-DXd for Trastuzumab-Deruxtecan in the article (lines 80, 467 and 682). Could you please clarify your comment?
11) "Line 480: three references ..... only one reference is written #22"
My apologies, it was a mistake on my part. This paragraph has been shortened for greater clarity (lines 472-490 instead of lines 491-521).
12) "Line 600: "first anti-HER2 treatments" ..... In the introduction section this therapy is called "conventional anti-HER2 therapy". I suggest using the same wording."
That is right, it's been modified (lines 639-640).
13) "The Discussion section, although too long, lacks the paragraph "limitations/restrictions of the study". The small sample size, lack of correlation with response to anti-HER2 therapy, conventional and/or ADCs, and endocrine therapy need to be highlighted. Data on outcome in relation to active pathways would provide a more comprehensive insight into the clinical relevance of the findings. This is important because although the new generation of anti-HER2 therapies target the HER2 protein, the by-stander effect they induce plays an important (crucial?) role. Therefore, the present work could be the backbone for further research in this area."
Thank you for your comments.
Indeed, you will find a new paragraph on the limitations and restrictions of the study at the end of the discussion (lines 658-665).
The size of the discussion has been shortened in terms of clinicopathological data, and some data less relevant to the main aim of the study have been removed (notably data on DE carcinomas, to focus on H2L carcinomas). All these modifications are tracked in the manuscript (deleted passages are crossed out).
You will find attached the new version of the manuscript, taking into account the comments of all the reviewers.

Round 2
Reviewer 2 Report
Comments and Suggestions for Authors
Dear authors,
The manuscript has been greatly improved and the current version is much easier to read and understand. I also have two minor comments:
1. Line 466: replace T-Dxd with T-DXd
2. Line 402: a typo: separate "a" from "similar".
I have no other comments.
Great work.